# Assessing the Measurement Invariance of the Human–Computer Trust Scale

**Gabriela Beltrão** *, **Sonia Sousa** and **David Lamas**

School of Digital Technologies, Tallinn University, 10120 Tallinn, Estonia; scs@tlu.ee (S.S.); drl@tlu.ee (D.L.)
* Correspondence: gbeltrao@tlu.ee

**Abstract:** Trust in technology is a topic of growing importance in Human–Computer Interaction due to the growing impact of systems on daily lives. However, limited attention has been paid to how one's national culture shapes their propensity to trust. This study addresses an existing gap in trust in technology research by advancing towards a more accurate tool for quantitatively measuring propensity to trust across different contexts. We specifically evaluate the psychometric properties of the human–computer trust scale (HCTS) in Brazil, Singapore, Malaysia, Estonia, and Mongolia. To accomplish this, we used the Measurement Invariance of Composite Models (MICOM), a procedure that examines the equivalency of the instrument's psychometric properties across different groups. Our results highlight the importance of rigorous validation processes when applying psychometric instruments in cross-cultural contexts, offering insights into the differences between the countries investigated and the procedure's potential to investigate trust across different groups.

**Keywords:** trust in technology; measurement invariance; trust assessment methods

## 1. Introduction

As technology increasingly permeates nearly every aspect of modern life, understanding how individuals trust systems has become critical in Human–Computer Interaction (HCI). Despite the growing importance of trust in technology, there has been limited attention toward validating trust assessment tools, as the rapid pace of technological change often conflicts with the time-intensive process of instrument development and validation. However, it is crucial to ensure that these tools function consistently across different demographic groups to enable meaningful cross-cultural analyses.

This study seeks to address this issue by advancing the Human–Computer Trust Scale (HCTS), a trust assessment instrument [1], by critically examining its applicability across Brazil, Singapore, Malaysia, Estonia, and Mongolia. We use Measurement Invariance of Composite Models (MICOM), a statistical procedure within Partial Least Squares Structural Equation Modeling (PLS-SEM), to assess whether the HCTS demonstrates consistent psychometric properties across the five countries [2–4]. Our guiding research question is whether measurement invariance can be established for the HCTS across the five countries with distinct cultural dimensions [5].

Additionally, despite a growing body of research on trust in technology [6,7], limited attention has been paid to how one's national culture shapes their propensity to trust technological systems. This article builds on the evidence that national culture moderates trust in technology [8] and focuses on providing a more rigorous assessment of the HCTS's psychometric properties. By exploring these dynamics, we provide insights into measuring propensity to trust and offer methodological guidance for applying MICOM in HCI research.

Thus, our main research question is complemented by our secondary goal, which is to explore if and how MICOM analysis can help identify differences in trust in technology between the countries.

This paper begins with an overview of trust in technology, the conceptual grounding of the HCTS, and the impact of national culture on trust. We then outline our data collection and analysis procedures, followed by a detailed account of the MICOM procedure and results. In the discussion, we integrate our findings to address the research questions and provide practical insights for conducting trust assessments across countries. By addressing these questions, our study enhances the validity of the HCTS across cultures and presents a methodological approach for future HCI research.

## 2. Theoretical Background

### 2.1. Trust in Technology

Trust is a fundamental mechanism enabling individuals to manage uncertainty in social interactions [9]. It plays a crucial role across various dimensions of life, from personal relationships to broader societal engagements [10]. When we look into human–technology interactions, these characteristics are, to some extent, similar. From an HCI perspective, trust in technology also involves rational and emotional responses [11], potentially improving individuals' acceptance, sustained usage, and satisfaction when interacting with technology [12].

With the rapid advancement of technology in recent decades, people now encounter new ways to interact with digital systems, introducing new layers of complexity to trust. These complexities arise from unclear perceptions of technological characteristics and the indirect factors that influence trust within human–technology relationships [13]. Consequently, measuring trust in technology has become both more complicated and important. For those reasons, our goal is to contribute to advancing the validation of an existing psychometric instrument designed to assess trust in technology.

Trust as an intention is different but is an antecedent to behavior [14]. We follow the conceptual groundings of the HCTS and adopt Mayer's [15] trust conceptualization, which explains it as one's willingness to be vulnerable based on the expectation that the other party will perform a particular action. This definition focuses on trust as a propensity and highlights that this intention is based on individuals' subjective assessment of the trustee—in this case, technology. It also points out that the propensity to trust depends on the situation, implying that there is some degree of uncertainty or risk for the trustor [16].

Trust in technology can also be approached as a behavior, where it is defined as an attitude that reflects an individual's belief that a system will help them achieve their goals in situations of uncertainty [17]. This concept differs from propensity, as it is characterized as an attitude that mediates interactions. While the two approaches are complementary and relevant, they entail different measures and yield distinct outcomes. We focus on *propensity*, which is more closely related to culturally shaped inclinations and is suitable for studying emerging technologies or hypothetical scenarios, thereby addressing an important aspect of HCI.

### 2.2. Human–Computer Trust Scale (HCTS)

The Human–Computer Trust Scale (HCTS) is a psychometric instrument designed to measure individuals' propensity to trust technology [18]. It is based on the Human–Computer Trust Model (HCTM), a theoretically grounded model of individuals' trust formation that approaches trust in technology through a socio-technical lens [19], that is, recognizing technology as embedded within social and organizational contexts.

The model and scale are derived from previous research on trust in technology, which examines the dynamics of trust formation in online relationships [14], as well as definitions

and measures that examine trust-related attributes within the technological artifact rather than in the associated people or organizations [16]. Although these cannot be entirely disassociated, they imply different objects.

The HCTS focuses on assessing trust propensity, that is, individuals' predispositions to trusting behaviors [20,21]. The direct translation of disposition to behavior is complex, as it is shaped by cognitive, social, and contextual influences [22,23]. However, understanding trust propensity is essential for anticipating interaction challenges [24].

The scale has been applied to measure trust predisposition in various technology use contexts, including e-voting, voice assistants, future scenarios [1], human–robot interaction, and messaging applications [25].

We use the revised HCTS [25], which includes one additional construct to the original scale [1] to better account for the characteristics of AI, as demonstrated in Figure 1. The model conceptualizes trust as the combination of individuals' perceptions based on four constructs:

- **Competence (COM)** refers to a system's capability to perform its intended tasks effectively through appropriate features and functionalities, meeting expectations. The concept and items are based on notions from Mcknight and colleagues [16,26], who draw from the idea of the system's functionality in online interactions, such as e-commerce.

- **Benevolence (BEN)** refers to the technology acting in the trustor's best interest, even when there is no obligation or reward to act in such a manner [14]. In practical terms, it translates into the system's ability to provide adequate support to help users achieve their interaction goals [1]. The items are based on the work of Bhattacherjee [14], who focuses on trust in online firms.

- **Perceived Risk (PR)** refers to individuals' subjective assessment of the potential consequences of adverse incidents when interacting with technology. It draws on notions from [27], who define Perceived Risk as a subjective belief in the possibility of loss due to potential opportunistic behaviors by online sellers, such as fraud or breaking their commitments. This construct is derived from notions of willingness, that is, the notion of the individual being open to relying on the technology [27], and honesty [26], the notion that the trustee will act with integrity. These constructs were initially included in the HCTM [18] but later combined and reformulated as Perceived Risk [1], a higher-order concept encompassing these notions.

- **Structural Assurance (SA)** refers to the belief that reliable legal, contractual, or physical mechanisms are in place to support and secure technology use. This construct was removed after the scale's initial validations as part of the conceptual model [18], but was re-introduced to address a gap in the scale when targeting AI-based technologies [REF Beltrao]. It was initially conceptualized as part of "institution-based trust", part of the structure in which the technology is used and not an individual's disposition [26]. Nevertheless, considering the development of technology in recent years and the fuzzy legal landscape of AI-based systems, we approach Structural Assurance from the perspective of the trustor's perception of it, regardless of the actual protection offered to them.

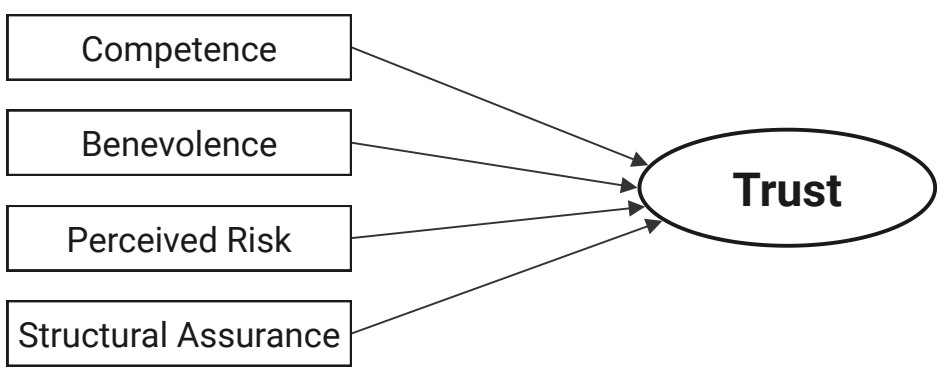

**Figure 1.** HCTS conceptual model.

*2.3. Differences in Trust Across Countries*

We approach trust from a socio-technical perspective [28], aiming to better understand the connection between social structures and norms that shape trust at a collective level. This understanding is fundamental in the current context of technology adoption, where technological solutions are created in one place and rapidly adopted across various locations. However, this often happens without understanding their potential impacts on the respective populations. Both too little or too much trust can negatively impact the interactions with technology and harm individuals [29].

Research shows that an individual's national culture can influence their tendency to trust others in interpersonal relationships [30,31]. Similarly, in HCI, studies have demonstrated that national culture can shape individuals' trust in technology [32–36]. Nevertheless, most studies are case-specific and allow only limited generalization. Thus, while there is a general agreement that national culture and trust in technology are connected, there is no clear understanding of this relationship.

Along the same lines, national culture can influence how trust in technology is measured. Variations in shared values among individuals can shape their perceptions of trust, which affects the effectiveness of psychometric instruments used to assess this concept. Therefore, ultimately, this study aims to contribute to understanding the relationship between individuals' national culture and their trust in technology.

This study explores differences between countries through the cultural dimensions framework, by Hofstede and colleagues [37]. This model is widely adopted for investigating cultural differences at the national level, positing that national cultures influence patterns of thinking, feeling, and behaving that are shared and passed down among members of the same nation. The authors identify six dimensions that shape culture, some of which may impact the citizens' tendency to trust one another. We acknowledge criticisms of the model regarding sample representativeness, universality, and relevance [38,39]; and the existence of other models for comparing cultures, such as GLOBE [40] or Schwartz's theory of basic values [41]. Nonetheless, Hofstede's work remains a comprehensive framework that enables practical cross-national comparisons, especially at more general levels, as intended by this study.

The countries included in the study are Brazil, Singapore, Malaysia, Estonia, and Mongolia. They were selected primarily due to their differences in cultural dimensions found to be related to trust, namely Power Distance, Individualism, and Uncertainty Avoidance:

- **Power Distance** refers to how the individuals of a society accept the distribution of power, shaping their view on how individuals in different power positions should interact [37]. In societies with higher Power Distance, individuals are more likely to recognize authority and hierarchical structures, so there is generally a higher trust in

authority figures. Conversely, a lower Power Distance leads to less centralized power and flatter structures [42–44]. It has been demonstrated that societies with a higher Power Distance are less open to accepting and adopting new technologies [45], which is interrelated to their propensity to trust these systems.

- **Collectivism–Individualism** describes the extent to which individuals are integrated into groups, with collectivist cultures emphasizing group harmony and cohesion over personal autonomy and individual achievements [5]. Yamagishi and Yamagishi [30] first identified the impact of collectivism on trust, and subsequent research has consistently demonstrated a positive effect between collectivism and trust propensity [46]. Notably, in collectivist societies, trust is often stronger among in-group members, whereas in individualist cultures, it is more broadly distributed [30].

- **Uncertainty Avoidance** reflects the extent to which a society feels threatened by ambiguity and unstructured situations, as opposed to tolerating them [5]. Cultures with high Uncertainty Avoidance generally prefer structure, order, and clear rules. Research indicates that this cultural dimension influences trust propensity [43]; specifically, individuals from high Uncertainty Avoidance cultures are more likely to trust when interactions are governed by clear structures and guidelines [47].

Here, we underline the effect of specific dimensions on trust alone to facilitate understanding. It is crucial to notice that each country has specific characteristics shaped by the unique combination of all the dimensions. The complete comparison, including all six dimensions, can be found in Appendix A.

## 3. Methodology

Our study follows the steps recommended for the MICOM procedure in the literature [2,48,49] to explore individuals' national culture's influence on the HCTS. While the MICOM is at the core of our article, we first evaluated the measurement model to ensure its validity in the countries included and later analyzed the path coefficients to further explore the effect of national culture on the model's behavior. All procedures are detailed in this section.

The five studies were conducted independently but followed the same protocol in each country between 2023 and 2024. This study presented a stimulus and relied on participants' perceptions regarding the hypothetical implementation of such a system in their country. Facial Recognition Systems (FRSs) were chosen for this study because they have not yet been widely implemented anywhere. This allowed participants to form impressions of the technology in a more abstract manner, making it easier to compare their predispositions. In contrast, using an existing system could trigger past experiences that are context-specific and may vary significantly between countries, complicating the comparison of results.

The stimulus consisted of a 2 min long video featuring excerpts of FRSs implemented in China (Skynet) and England (Metropolitan Police of London). Each excerpt showcased real applications of the technology with slightly different approaches: Skynet focuses on how the system can enhance safety, while the Metropolitan Police emphasizes the importance of maintaining citizens' privacy. The objective of this stimulus was to help participants understand the potential benefits and risks associated with FRSs without being limited to a specific example.

The questionnaire included items on socio-demographic factors, including age, gender, and education level; items related to technology usage and access; and the HCTS. The HCTS was tailored to reflect the focus on FRSs, following the most recent version of the instrument, containing 11 items measured on a 5-point Likert scale [25], and three items measuring Trust for the PLS-SEM validation [1].

In all cases, the questionnaire was also made available in English, its originally validated version. Additionally, translated versions were provided by native speakers in the local languages: Portuguese in Brazil; Chinese, Tamil, and Malay in Singapore; Chinese and Malay in Malaysia; Estonian in Estonia; and Mongolian and Chinese in Mongolia. Participants were recruited through convenience sampling, with assistance from local institutions and the researchers' networks in each country. Data collection occurred online via LimeSurvey (https://www.limesurvey.org/ accessed 11 November 2024).

*Samples*

Participants were recruited using a convenience sampling strategy, which leveraged the authors' network while also striving to include a diverse sample. We chose convenience sampling because of resource limitations that made probability sampling impractical. Since this is a preliminary investigation aimed at validating the measurement instrument and identifying patterns, a convenience sample is appropriate. However, we emphasize that further studies should focus on generalizability, as our sample may not be fully balanced or representative of the broader populations.

The sample size (N) for each country considered in this study is as follows: Brazil = 133, Singapore = 109, Malaysia = 107, Estonia = 117, and Mongolia = 120, resulting in a total sample size of N = 586. A detailed description of the sample breakdown by gender and age range can be seen in Table 1.

**Table 1.** Summary of sample characteristics.

|  | **Brazil** | **Estonia** | **Malaysia** | **Mongolia** | **Singapore** | **Total** |
|---|---|---|---|---|---|---|
| **Total** | **133** | **117** | **107** | **120** | **109** | **586** |
| **Gender** |  |  |  |  |  |  |
| Female | 41 | 70 | 46 | 74 | 73 | 304 |
| Male | 89 | 44 | 59 | 46 | 36 | 274 |
| Other | 3 | 3 | 2 |  |  | 8 |
| **Age Range** |  |  |  |  |  |  |
| 17 or less | 3 |  |  | 2 |  | 5 |
| 18–24 | 93 | 14 | 76 | 53 | 2 | 238 |
| 25–34 | 28 | 36 | 11 | 32 | 32 | 139 |
| 35–44 | 4 | 41 | 16 | 23 | 45 | 129 |
| 45–54 | 5 | 22 | 3 | 9 | 17 | 56 |
| 55–64 |  | 4 | 1 |  | 8 | 13 |
| 65 or more |  |  |  | 1 | 5 | 6 |

## 4. Analysis

### 4.1. Measurement Model Evaluation

Before conducting the MICOM analysis, we assessed the measurement model for each sample. Although the HCTS has already been validated, this assessment is advisable before the MICOM to ensure that the properties of the constructs remain in the different contexts investigated [50].

Table 2 presents the HTCS questionnaire adopted for the assessment. It comprises the constructs Competence (COM), Benevolence (BEN), Perceived Risk (PR) (reversed), and Structural Assurance (SA). The content within brackets refers to the specific technology assessed.

**Table 2.** Instrument under assessment (HCTS).

| Construct | Item |
|---|---|
| COM1 | [Facial recognition systems] are competent and effective in [identifying dangerous individuals]. |
| COM2 | [Facial recognition systems] perform their role in [identifying potentially dangerous individuals] very well. |
| COM3 ** | [Facial recognition systems] have all the functionalities I would expect from [Artificial Intelligence]. |
| PR1 | There could be negative consequences when using [facial recognition systems]. |
| PR2 ** | I must be cautious when using [facial recognition systems]. |
| PR3 | It is risky to interact with [facial recognition systems]. |
| BEN1 | [Facial recognition systems] will act in my best interest. |
| BEN2 | [Facial recognition systems] will do their best to help me if I need help. |
| BEN3 | [Facial recognition systems] are interested in understanding my needs and preferences. |
| SA1 | I feel assured that legal and technological structures provided by the government protect me when using [facial recognition systems]. |
| SA2 | I can trust [facial recognition systems] because [Artificial Intelligence] systems are robust and safe. |
| TR1 * | I am willing to use [facial recognition systems]. |
| TR2 * | I can rely on [facial recognition systems] for [law enforcement]. |
| TR3 * | I can trust the outcomes of [facial recognition systems]. |

* Trust (TR) items were used to assess the measurement and structural model using PLS-SEM. ** Items removed due to low loadings.

Our assessment of the measurement models focused on the measures' internal consistency, reliability, convergent validity, and discriminant validity. The results showed considerable variability in the measures' reliability and validity.

Reliability values refer to how well the indicator reflects the latent construct. Values above 0.5 are considered adequate [3]. All the samples presented at least one problematic item in all cases in the construct COM or PR. The most critical issues were in the construct COM3 for the samples of Singapore and Estonia, and the construct PR2 for Mongolia and Malaysia.

Next, we looked at the average variance extracted (AVE), which represents how well the indicators explain the constructs. Following the problems with the item's reliability, PR was below the 0.5 threshold for Mongolia. Finally, we considered composite reliability (CR), which assesses the internal consistency of the indicators. The problems remained, with values for PR below the necessary thresholds of 0.7 for Malaysia and Mongolia. Addressing these issues, we removed the constructs with the lowest loadings: COM3 and PR2.

The model without these items yielded better results, with only one loading below the 0.7 threshold for the Malaysia sample (PR3). However, it did not affect the AVE and CR, which were adequate for this version of the model. The results can be seen in Table 3. The results for the original model are available in Appendix A (Table A1).

**Table 3.** Summary of measurement model evaluation results—refined model.

| Item | Construct | Reliability (>0.5) | | | | | AVE (>0.5) | | | | | CR (>0.7) | | | | |
|---|---|---|---|---|---|---|---|---|---|---|---|---|---|---|---|---|
| | | BR | SIN | MAL | EE | MON | BR | SIN | MAL | EE | MON | BR | SIN | MAL | EE | MON |
| BEN1 | | 0.719 | 0.752 | 0.728 | 0.789 | 0.726 | | | | | | | | | | |
| BEN2 | BEN | 0.837 | 0.750 | 0.721 | 0.794 | 0.733 | 0.771 | 0.722 | 0.661 | 0.744 | 0.693 | 0.855 | 0.808 | 0.787 | 0.864 | 0.788 |
| BEN3 | | 0.753 | 0.663 | 0.534 | 0.648 | 0.619 | | | | | | | | | | |
| COM1 | COM | 0.575 | 0.776 | 0.734 | 0.773 | 0.714 | 0.707 | 0.815 | 0.767 | 0.794 | 0.784 | 0.704 | 0.798 | 0.708 | 0.747 | 0.789 |
| COM2 | | 0.841 | 0.852 | 0.801 | 0.815 | 0.854 | | | | | | | | | | |
| PR1 | PR | 0.776 | 0.591 | 0.912 | 0.745 | 0.821 | 0.800 | 0.733 | 0.651 | 0.790 | 0.756 | 0.757 | 0.820 | 0.886 | 0.761 | 0.716 |
| PR3 | | 0.823 | 0.874 | 0.391 ** | 0.835 | 0.691 | | | | | | | | | | |
| SA1 | SA | 0.841 | 0.794 | 0.828 | 0.850 | 0.619 | 0.852 | 0.811 | 0.846 | 0.827 | 0.735 | 0.831 | 0.773 | 0.828 | 0.802 | 0.756 |
| SA2 | | 0.865 | 0.830 | 0.865 | 0.803 | 0.852 | | | | | | | | | | |
| TR1 | | 0.659 | 0.623 | 0.699 | 0.738 | 0.843 | | | | | | | | | | |
| TR2 | TR | 0.796 | 0.787 | 0.656 | 0.792 | 0.805 | 0.756 | 0.738 | 0.701 | 0.775 | 0.855 | 0.851 | 0.834 | 0.814 | 0.856 | 0.920 |
| TR3 | | 0.812 | 0.803 | 0.748 | 0.796 | 0.918 | | | | | | | | | | |

** Values below the threshold. AVE = average variance extracted; CR = coefficient of reliability; BR = Brazil, SIN = Singapore, MAL = Malaysia, EE = Estonia, MON = Mongolia.

Next, we assessed the discriminant validity (DV) for the refined model using the Fornell–Larcker criterion. This criterion evaluates the correlations between each item and the constructs in the model. In all samples, each construct's highest correlation was with its own construct, demonstrating that there are no issues with DV. The table with the DV

results is available in the Appendix B (Table A2). The highest values per construct per country are highlighted to facilitate interpretation. Finally, we looked at R², representing the explained variance of the endogenous construct (Trust) by the exogenous constructs (COM, BEN, PR, and SA). The values were as follows: Brazil = 0.625, Singapore = 0.589, Malaysia = 0.510, Estonia = 0.682, and Mongolia = 0.736; all were considered adequate in our domain of study.

After removing two items, all the samples met most of the assessment criteria for reliability, convergent, and discriminant validity. Thus, we proceeded with the MICOM using the adjusted model with nine items. The problems encountered in the measurement model evaluation are further addressed in the Section 5.

### 4.2. Measurement of Invariance Assessment (MICOM)

Next, we proceeded with the MICOM. The procedure is composed of three steps, which should be followed if the previous criteria have been met: (1) configural invariance, (2) compositional invariance, and (3) equal mean values and variances [48].

### 4.2.1. Configural Invariance

The first step refers to the assessment of the conceptual model structure, requiring (1) identical indicators per measurement model, (2) identical data treatment, and (3) identical algorithm settings [4].

Following the studies' design, the scale was implemented with the same items and under the same protocol in all the countries. The scale was administered in English and the local languages, following a back-translation process by native speakers. Our group-specific model estimations draw on identical algorithm settings, and due to the measurement model evaluation and adjustments in the previous step of the analysis, we can also consider that the PLS path model setups are equal across the three countries. **Thus, configural invariance is established**.

### 4.2.2. Compositional Invariance

Compositional invariance assesses if the relationships between indicators and the composite constructs are similar across groups. This step is required to ensure that the constructs are formed in the same way across the countries, which is necessary for comparing results between them [4].

This procedure was performed in paired comparisons. Since we had five countries, we had a total of 10 comparisons. We ran the permutation procedure with 1000 permutations and a 5% significance level for each paired comparison.

To assess compositional invariance, we compared the original correlations of composite scores (C) with those generated from the permutation test (Cu). If C is higher than the 5% threshold of Cu, also reflected in non-significant $p$-values ($p > 0.05$), compositional invariance is established.

A permutation $p$-value > 0.05 means that the difference in the construct's composition is not significantly different, so their results can be compared. Table 4 presents the p-values of the comparisons to facilitate the overview of the results. **Only three pairs from our samples achieved full compositional invariance: Singapore vs. Brazil, Singapore vs. Estonia, and Mongolia vs. Malaysia**. All other paired comparisons had between one and three constructs violating compositional invariance. The violation most commonly happened for the endogenous construct (Trust). The values in bold represent the cases in which compositional invariance was achieved, indicating that they can be compared. For some comparisons, the difference was significant only for Trust but close to the 0.05 threshold. Similarly, some comparisons had only one significantly different construct.

**Table 4.** Compositional invariance significance test summary.

| | BRxSIN | BRxMAL | BRxEE | BRxMON | SINxMAL | SINxEE | SINxMON | MALxEE | MALxMON | EExMON |
|---|---|---|---|---|---|---|---|---|---|---|
| | | | | | **Permutation *p*-Value** | | | | | |
| BEN | **0.941** | 0.011 | 0.019 | **0.597** | **0.123** | **0.139** | **0.712** | **0.893** | **0.446** | **0.457** |
| COM | **0.325** | **0.378** | **0.179** | **0.675** | **0.785** | **0.612** | **0.682** | **0.869** | **0.659** | **0.468** |
| PR | **0.155** | 0.009 | **0.593** | **0.392** | **0.056** | **0.244** | **0.265** | 0.003 | **0.576** | **0.227** |
| SA | **0.937** | **0.850** | **0.106** | **0.157** | **0.894** | **0.081** | **0.217** | **0.058** | **0.231** | 0.003 |
| Trust | **0.833** | 0.002 | **0.079** | 0.045 | 0.036 | **0.243** | 0.047 | 0.008 | **0.089** | **0.088** |
| | ✓ | | | | | | ✓ | | ✓ | |

Values in bold represent *p* > 0.05, indicating that the difference in the construct's composition is **not** significantly different. ✓ represents the cases where compositional invariance was achieved. BR = Brazil, SIN = Singapore, MAL = Malaysia, EE = Estonia, MON = Mongolia.

Although full compositional invariance was not achieved in most cases, we proceeded with further analyses to understand the differences between groups in more depth. This decision is based on the fact that the HCTS has already been validated and variations in invariance have been observed across the samples. Furthermore, it aligns with the exploratory nature of the MICOM, allowing us to further examine whether these problems are reflected in the analysis of equal means and variances.

*4.3. Equal Mean Values and Variances*

The final step of MICOM evaluates whether the mean values and variances of the constructs are equal across different groups [4]. Equal means indicate that the groups have similar tendencies for each construct, and equal variances imply similar dispersion. If the means and variances are considered equal, full measurement invariance is achieved, and the data from different groups can be pooled. It also means that any differences in path coefficients can be interpreted confidently rather than attributed to measurement variability.

Results are calculated similarly to compositional invariance, with *p*-values > 0.05 indicating that the differences are not significantly different, and that the results can be compared across the groups. Within our samples, no paired comparison presented equal composite means, and the similarities varied considerably between the pairs. Regarding variance, Brazil vs. Estonia, Singapore vs. Malaysia, and Singapore vs. Estonia had equal variances for all constructs. The number of constructs achieving measurement invariance in other paired comparisons varied considerably. Tables 5 and 6 present overviews of the results, with values in bold representing the cases in which the conditions were satisfied.

To answer our research question, **measurement invariance was not achieved for the HCTS in most cases**. Partial measurement invariance was achieved for the pairs Singapore vs. Brazil, Singapore vs. Estonia, and Mongolia vs. Malaysia, indicating that only in these cases can the HCTS results be compared, but not pooled.

**Table 5.** Summary of mean value significance tests.

| | BRxSIN | BRxMAL | BRxEE | BRxMON | SINxMAL | SINxEE | SINxMON | MALxEE | MALxMON | EExMON |
|---|---|---|---|---|---|---|---|---|---|---|
| | | | | | **Permutation *p*-Value** | | | | | |
| BEN | 0.011 | **0.429** | 0.000 | **0.078** | 0.000 | **0.321** | 0.000 | 0.000 | **0.437** | 0.000 |
| COM | **0.335** | **0.230** | **0.101** | **0.105** | 0.050 | **0.569** | **0.380** | 0.009 | 0.011 | **0.761** |
| PR | **0.053** | **0.093** | 0.000 | **0.304** | **0.781** | 0.040 | 0.002 | 0.013 | 0.006 | 0.000 |
| SA | 0.001 | 0.000 | **0.060** | 0.000 | **0.064** | 0.000 | 0.004 | 0.000 | **0.265** | 0.000 |
| Trust | **0.462** | **0.779** | 0.000 | **0.526** | **0.241** | 0.002 | **0.210** | 0.000 | **0.776** | 0.000 |

Values in bold represent *p* > 0.05, indicating that the difference in the construct's composition is **not** significantly different. BR = Brazil, SIN = Singapore, MAL = Malaysia, EE = Estonia, MON = Mongolia.

**Table 6.** Summary of variance significance tests.

| | BRxSIN | BRxMAL | BRxEE | BRxMON | SINxMAL | SINxEE | SINxMON | MALxEE | MALxMON | EExMON |
|---|---|---|---|---|---|---|---|---|---|---|
| | | | | | Permutation *p*-Value | | | | | |
| BEN | **0.056** | 0.021 | **0.157** | **0.518** | **0.651** | **0.585** | 0.011 | **0.273** | 0.006 | 0.023 |
| COM | **0.653** | **0.612** | **0.948** | 0.021 | **0.954** | **0.763** | 0.015 | **0.721** | 0.010 | 0.021 |
| PR | 0.004 | 0.017 | **0.157** | **0.955** | **0.830** | **0.196** | 0.032 | **0.439** | 0.037 | **0.170** |
| SA | **0.197** | **0.696** | **0.765** | **0.339** | **0.156** | **0.379** | **0.052** | **0.414** | **0.557** | **0.187** |
| Trust | **0.057** | 0.001 | **0.396** | **0.193** | **0.291** | **0.281** | 0.002 | 0.036 | 0.000 | 0.050 |
| | | | ✓ | | ✓ | ✓ | | | | |

Values in bold represent *p* > 0.05, indicating that the difference in the construct's composition is **not** significantly different. ✓ represents the cases where compositional invariance was achieved. BR = Brazil, SIN = Singapore, MAL = Malaysia, EE = Estonia, MON = Mongolia.

If measurement invariance is not achieved, comparing path coefficients can be problematic because differences might be due to the measurement varying across groups rather than genuine differences in the relationships between constructs. Nevertheless, based on our exploratory intentions, we analyze the path coefficients. This analysis focuses on cautious relative comparisons, taking into account the results of the MICOM.

### 4.4. Path Coefficients

We proceed with the path coefficients analysis, considering that our comparisons reached mostly partial compositional invariance. Table 7 presents the path coefficients per country. There, it can be observed that certain countries have higher similarities. For instance, based on the weights of the constructs BEN and COM, it is possible to identify two groups: Brazil, Singapore, and Mongolia, with a higher weight of BEN over COM, and Singapore and Estonia, with an inverse proportion of these constructs' weights. Additionally, Figure 2 is included to facilitate the visualization of the results.

**Table 7.** Path coefficients per country.

| | BR | SIN | MAL | EE | MON |
|---|---|---|---|---|---|
| **BEN → Trust** | 0.388 | 0.133 | 0.400 | 0.162 | 0.346 |
| **COM → Trust** | 0.145 | 0.414 | 0.128 | 0.330 | 0.220 |
| **PR → Trust** | −0.243 | −0.134 | −0.066 | −0.116 | −0.155 |
| **SA → Trust** | 0.244 | 0.298 | 0.293 | 0.409 | 0.404 |

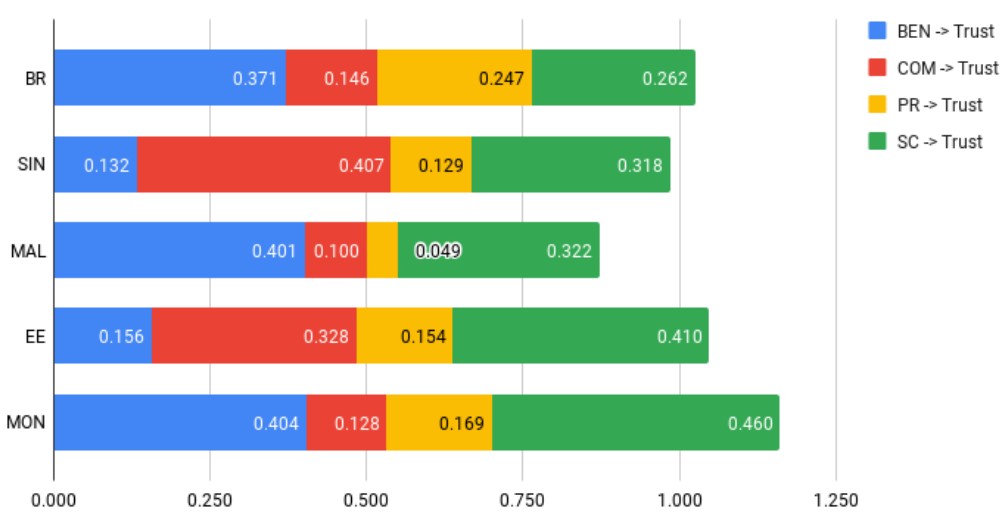

**Figure 2.** Path coefficients per country.

To evaluate the significance of the differences, we ran a bootstrapping analysis with 5000 subsamples at a 0.05 significance level. The analysis revealed that the difference

was only significant in a few comparisons and on more than two constructs per case. The bootstrapping table with complete results is available in Appendix C (Table A3).

Therefore, although we identified differences between the groups, the lack of statistical significance indicates that the findings must be interpreted cautiously. Figure 3 presents the summary of the procedure adopted, highlighting the outcomes of each step. The next section provides a complete reflection of our findings.

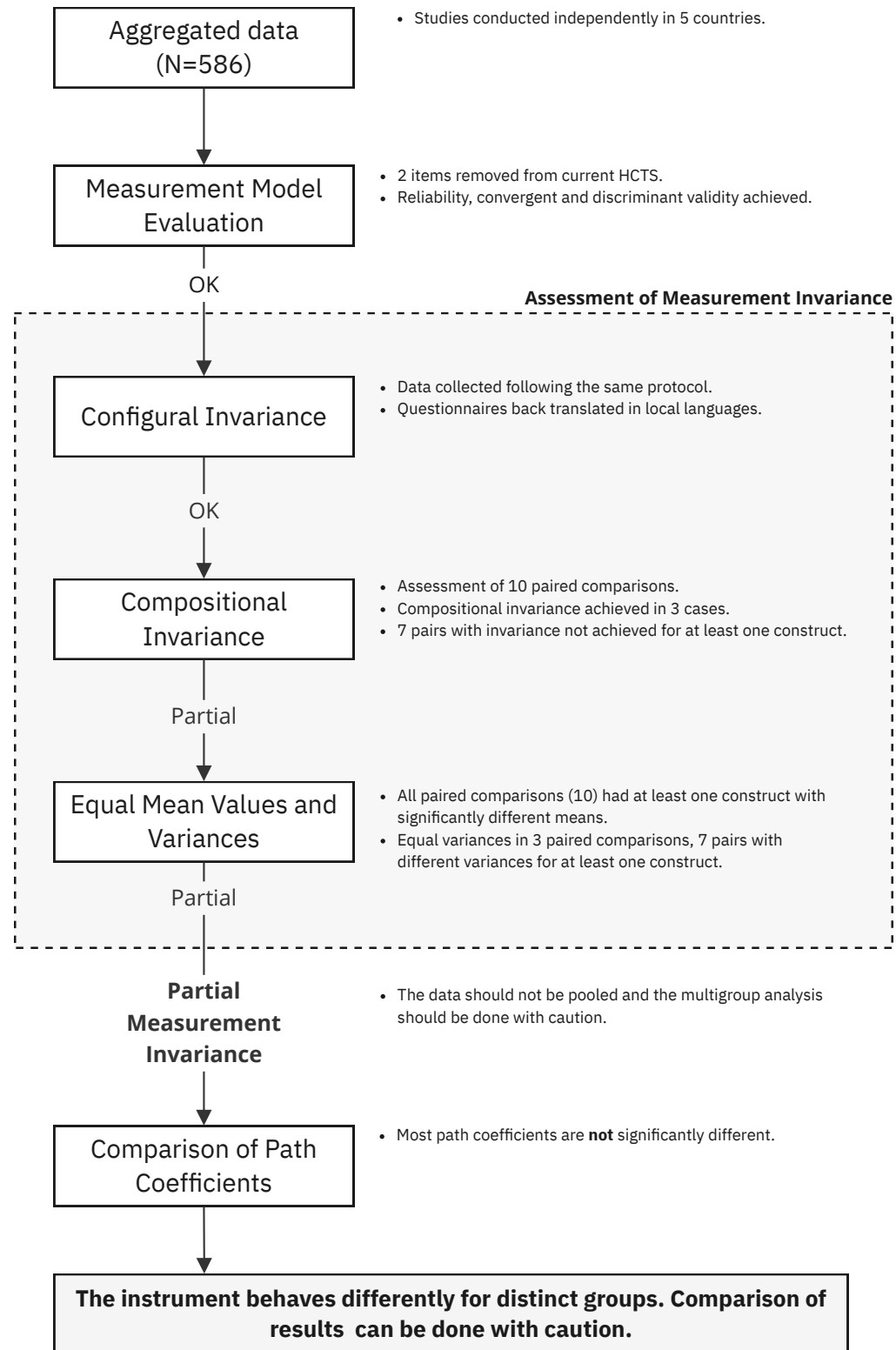

**Figure 3.** Summary of procedure and respective outcomes.

## 5. Discussion

The HCTS is a validated instrument for assessing trust in technology, which has been validated and subsequently applied in various contexts with distinct groups of participants. Although researchers have been careful with the application of the scale, limited attention has been paid to the effects of the context of the evaluation on the instrument's functioning. The MICOM results demonstrated the procedure's potential for a deeper understanding of the instrument and the potential for exploring the groups investigated. In this section, we discuss the implications of our findings, aiming to assist other researchers and practitioners in HCI in implementing the procedure.

First, the measurement model evaluation, a prerequisite for the MICOM, revealed that some constructs were problematic in particular samples, pointing towards a review of the scale [25] to ensure higher adequacy in different contexts—in this case, countries. The items with loading inadequacies varied according to the samples, providing initial indications that the scale behaves differently in each country, and not necessarily that there are structural problems with the scale [50], which we further explored with the MICOM. After the two items with persistent issues, COM3 and PR2, were removed, the scale's reliability and convergent validity were improved, and adequate DV and $R^2$ were achieved for all samples, fulfilling the prerequisites for running the MICOM. Furthermore, it supported the assumption that there were problems with these items. As both constructs (COM and PR) remained with two items each, the removal improved the scale's measurement quality without compromising the constructs' conceptual meaning.

Thus, the first contribution of this study is the advancement of the HCTS [1], building on the revised version [8]. The revised scale is available in Appendix D Table A4, and can more consistently be used across different national groups. Insights into the behavior of the HCTS and of the countries investigated are described next.

### 5.1. Measurement of Invariance Assessment (MICOM)

The first step of the MICOM analysis, configural invariance, assessed the conceptual model structure. This condition was met because the data were collected following the same protocol and analyzed using the same procedures.

The next step, the analysis of compositional invariance, is a requisite for confidently comparing the results between groups. Our empirical research included five samples, so we ran ten paired comparisons. This analysis revealed that 3 out of 10 pairs have significantly different compositions of at least one construct. As shown in Table 4, the invariance was most commonly not achieved for the endogenous construct (Trust). This finding suggests that the differences observed primarily relate to how the constructs affect trust rather than how the items compose the exogenous constructs. From a broader perspective, this result also points to variations in trust formation between the groups compared [8], rather than distinct interpretations of single items.

This result has more serious implications for the HCTS, as the lack of compositional variance indicates that the constructs are formed differently across groups, and thus, their comparison can be misleading [4], as the differences may be a result of differences in the measurement model. **In practical terms, this implies that comparisons between countries can only be confidently conducted between the pairs in which compositional invariance was achieved.** Figure 4 summarizes the paired comparisons. The pairs with check marks (✓) achieved compositional invariance. The others have partial compositional invariance, and the numbers in the cells reflect the amount of constructs that do not satisfy the condition. For the pairs with partial compositional invariance, only the constructs that satisfy the condition should be compared, as per Table 4.

|  | SING | MAL | EE | MON |
|---|---|---|---|---|
| **BR** | ✓ | 3 | 1 | 1 |
| **SING** |  | 1 | ✓ | 2 |
| **MAL** |  |  | 2 | ✓ |
| **EE** |  |  |  | 3 |

**Figure 4.** Summary of compositional invariance results.

According to MICOM's guidelines, the third step of the analysis should only be performed if compositional invariance is achieved [4], which was not the case in our study. Most pairs did not reach compositional invariance, as shown in Table 4. However, three pairs did not meet the criteria but had a single borderline variance value for Trust, while two others had variances for a single exogenous construct each. Considering that these results are not so far from meeting the criteria in most cases, we chose to proceed with the analysis. Nevertheless, we stress that this decision deviates from a rigorous MICOM procedure and is justified by our aim to provide further exploratory insights into our work.

In the third and final MICOM step, the assessment of equal means and variances, no pair had equal composite mean values, but three pairs had equal variances. The fact that none of the group pairs had equal mean values indicates that the average scores on the constructs differ between the groups. This suggests that individuals from different countries likely perceive trust in technology differently. For the three pairs that exhibited equal variances (Brazil vs. Mongolia, Singapore vs. Malaysia, and Singapore vs. Estonia), the consistency or spread of responses within those groups is similar. More specifically, the only pair that satisfied the conditions for compositional invariance and equal variances was Singapore vs. Estonia. These results imply that while significant differences exist between the countries, some comparisons remain valid, particularly regarding the relationships between constructs.

Thus, we can assume that the scale has partial compositional invariance across the countries investigated, with the invariance varying between the countries compared. This means that the data cannot be pooled, and comparing the results between countries is feasible with caution [4],considering which specific indicators or constructs differ across the analyzed groups.

Although our results are limited to five countries, the findings provide evidence that it is necessary to further evaluate the behavior of the psychometric instrument (HCTS or others) before making cross-country comparisons, as the differences between the populations can affect the relationship between endogenous and exogenous constructs. In practice, it means that the comparison of results between groups might embed the differences in their interpretation of the constructs.

Interestingly, the study outcomes also enabled us to explore these differences further. The results helped us identify areas where the instrument behaves consistently or inconsistently, serving as a basis for understanding the disparities between groups and improving the accuracy of our trust assessment results. Building on this, we examined the results for each country.

*5.2. Effects of National Culture on the HCTS*

The path coefficients (Table 7 and Figure 2) point towards the existence of two main groups among the countries. The first includes Brazil, Malaysia, and Mongolia, in which COM has a considerably higher weight than BEN in shaping Trust. Additionally, Malaysia and Mongolia have similar proportions of PR and SC.

Another group can be identified between Singapore and Estonia, where the weights of all constructs follow similar proportions. Most notably, BEN and COM have inverse weights compared to the other group. The bootstrapping results revealed that the differences are not statistically significant in most cases, but this might have been caused by the lack of full measurement invariance, which can affect the validity of the estimations [3].

If analyzed with the MICOM results, we can see that the paired comparisons between Malaysia and Mongolia, and Singapore and Estonia, reached compositional invariance, and Brazil and Mongolia had a borderline result; and Singapore and Estonia have additionally equal variances. These findings suggest that the outcomes of the HCTS for these specific pairs can be compared with greater confidence. Notably, this interpretation prioritizes the triangulation of the methods over the strictness of the thresholds to explore how the procedures applied can be used to explore further differences between the groups, which can lead to practical insights. However, we emphasize that these findings are speculative and should be used to guide further investigations and not generalizations.

Returning to Hofstede's cultural dimensions model [37], we observe that it provides limited insight into the groupings. As illustrated in Figure 5, Singapore and Estonia, which show the highest similarities in how the model functions, have notably different scores for Power Distance (Estonia = 40, Singapore = 74) and Uncertainty Avoidance (Estonia = 60, Singapore = 8). While they also differ in Individualism (Estonia = 62, Singapore = 43), both nations have the highest scores among the analyzed countries. One hypothesis is that their higher levels of Individualism foster similar views on autonomy and privacy, which, in turn, may shape their understanding of trust in technology.

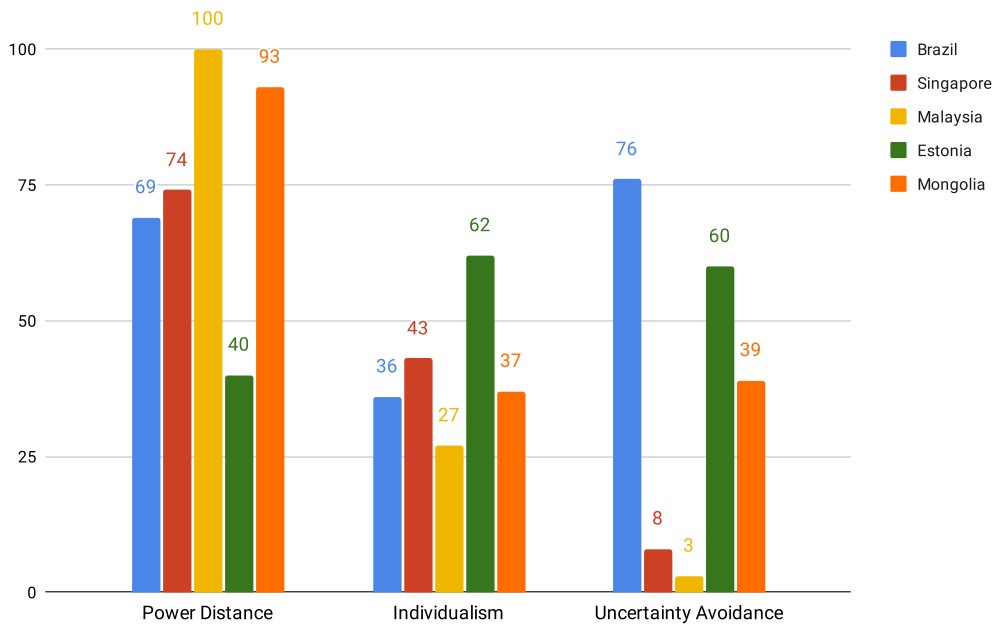

**Figure 5.** Country comparison based on the cultural dimensions model, presenting three dimensions.

Malaysia and Mongolia exhibit close values for Power Distance (Malaysia = 100, Mongolia = 93). This dimension might also be related to these populations' similar interpretations of trust in technology stemming from analogous views on authority and

their acceptance of it. In both cases, the technological object under discussion, FRSs, may have intensified these relationships. However, we stress that these interpretations are speculative, and further studies are needed to investigate these hypotheses, taking into account the unique characteristics of each country as defined by their cultural dimensions or by considering alternative models.

*5.3. Implications*

Through the implementation of the MICOM, this study demonstrates that the HCTS, an instrument to measure propensity trust in technology, can behave differently across countries. As such, the main implication of our study is that the comparison of such assessment results between countries may be misleading if these differences are not accounted for. The most rigorous solution to this issue is to follow the MICOM procedure before making such comparisons. However, as this may not be feasible, other strategies may be adopted to mitigate this issue, such as investigating the differences between the countries from qualitative-oriented approaches.

In addition, the results indicate that the understanding of trust in technology can vary significantly from one country to another, shaped by cultural perceptions and values. This is crucial for HCI because understanding these differences is essential for designing technologies that better align with users' expectations in various regions.

By highlighting how trust formation varies across cultures, this study also demonstrates how multilayered this topic is. While the focus of our study is to move towards more accurate assessments using the HCTS, the analysis also led to insights about the differences in the meaning of trust in technology for the groups. It is also noteworthy that we approached culture from a single perspective, from the national lens, which is one among numerous ways to investigate culture. Our results underscore the necessity for further cross-cultural investigations, following other approaches.

If we consider emerging discussions about reaching the optimal, not the highest, levels of trust [29], these findings imply that trust calibration mechanisms must be tailored, considering the varying expectations of system performance and reliability. For instance, in Brazil, Malaysia, and Mongolia, mechanisms focused on Benevolence [14], such as providing adequate support or fostering community involvement, could more strongly influence trust. Conversely, Competence plays a greater role in Estonia and Singapore, so demonstrating that the system meets high technical standards, has precision, and is reliable [26] would have more meaningful effects on trust.

Finally, our outcomes demonstrate the complex and contextual nature of trust in technology. While the insights are useful for designing systems that address different concerns, ethical considerations must be prioritized. Knowledge about differences in trust can improve interactions and empower individuals, but it can similarly be used to deceive them and exacerbate existing disparities. This is even more crucial when considering our object of study, FRSs, as this technology has strong social implications. This discussion is beyond the scope of our research, as here, FRSs were used merely as a prompt, and questions regarding their actual implementation require a much more detailed account. Nevertheless, we highlight that researchers and practitioners investigating FRSs, and more generally, trust in technology, must follow ethical practices and commit to respecting fundamental rights.

## 6. Conclusions

This study contributes to the field of trust in technology by advancing the cross-country validation of the HCTS. By employing the MICOM procedure, we assessed the psychometric properties of the HCTS across five culturally diverse countries: Brazil, Singapore, Malaysia, Estonia, and Mongolia. Our findings revealed partial measurement

invariance, indicating that while the instrument shows potential for cross-cultural applications, the results should be interpreted with caution because there are differences in how trust is understood and formed across these groups.

Our results underscore the importance of rigorous validation processes when applying psychometric instruments in cross-cultural contexts. Without such assessments, researchers may draw inaccurate conclusions about trust differences based on measurement variances rather than genuine disparities. The exploratory nature of our approach highlights the challenges of establishing full invariance and demonstrates the method's potential in identifying patterns among the groups.

Additionally, our outcomes provide evidence of the interplay between national culture and trust in technology, emphasizing the need for future research to move beyond national boundaries to explore other dimensions of culture. While we focused on a single case (facial recognition systems for law enforcement), the findings are also useful for reflecting on the interaction with other applications.

Overall, this study takes a step toward enhancing the robustness and applicability of the HCTS for cross-cultural research and understanding differences in trust in technology across countries. We expect our findings to guide further culturally sensitive research on trust in technology.

## 7. Limitations

This study has several limitations that should be acknowledged. First, the sample used for analysis is not representative of the population of each country, and differences in the sample characteristics, such as gender and age distributions, may have influenced the results, which limits the generalizability of the findings. Second, the MICOM-recommended thresholds were not strictly adhered to in all cases. This decision was guided by the intention to allow further exploration of the topic, but it may have affected the robustness of some findings. In addition, using countries as proxies for cultural background represents only a superficial dimension of individuals' cultural identity. Countries do not represent homogeneous cultural groups, and this approach may overlook important within-country variations. Future research should include more diverse and representative samples and explore additional dimensions of cultural differences to provide a more nuanced understanding of the interplay between culture and trust in technology.

**Author Contributions:** Conceptualization, G.B., S.S., and D.L.; methodology, G.B., S.S., and D.L.; validation, G.B.; formal analysis, G.B.; investigation, G.B.; data curation, G.B.; writing—original draft preparation, G.B.; writing—review and editing, D.L.; visualization, G.B.; supervision, S.S. and D.L.; project administration, S.S. and D.L.; funding acquisition, S.S. and D.L. All authors have read and agreed to the published version of the manuscript.

**Funding:** This work was supported by the project MARTINI, grant CHIST-ERA-21-OSNEM-004 (TKA22209), and the European Office of Aerospace Research and Development and US Air Force Office of Scientific Research: FA8655-22-1-7051.

**Institutional Review Board Statement:** All subjects gave their informed consent for inclusion before they participated in the study. The study was conducted according to the guidelines of the Declaration of Helsinki and approved by the Ethics Committee of Tallinn University (protocol code 20 and date of approval 14 June 2021).

**Informed Consent Statement:** Informed consent was obtained from all subjects involved in the study.

**Data Availability Statement:** The original data presented in the study are openly available in the Open Science Framework (OSF) at https://osf.io/xzamn/?view_only=90946a29e1d44f50b511239dcc3ed59d. (accessed on 21 April 2025).

**Conflicts of Interest:** The authors declare no conflicts of interest.

## Appendix A. Country Comparison

Figure A1 presents the complete comparison of Brazil, Singapore, Malaysia, Estonia, and Mongolia as per the country comparison tool, based on the cultural dimensions model [5]. The comparison was enabled through the online tool available at https://www.theculturefactor.com/country-comparison-tool (accessed on 21 April 2025).

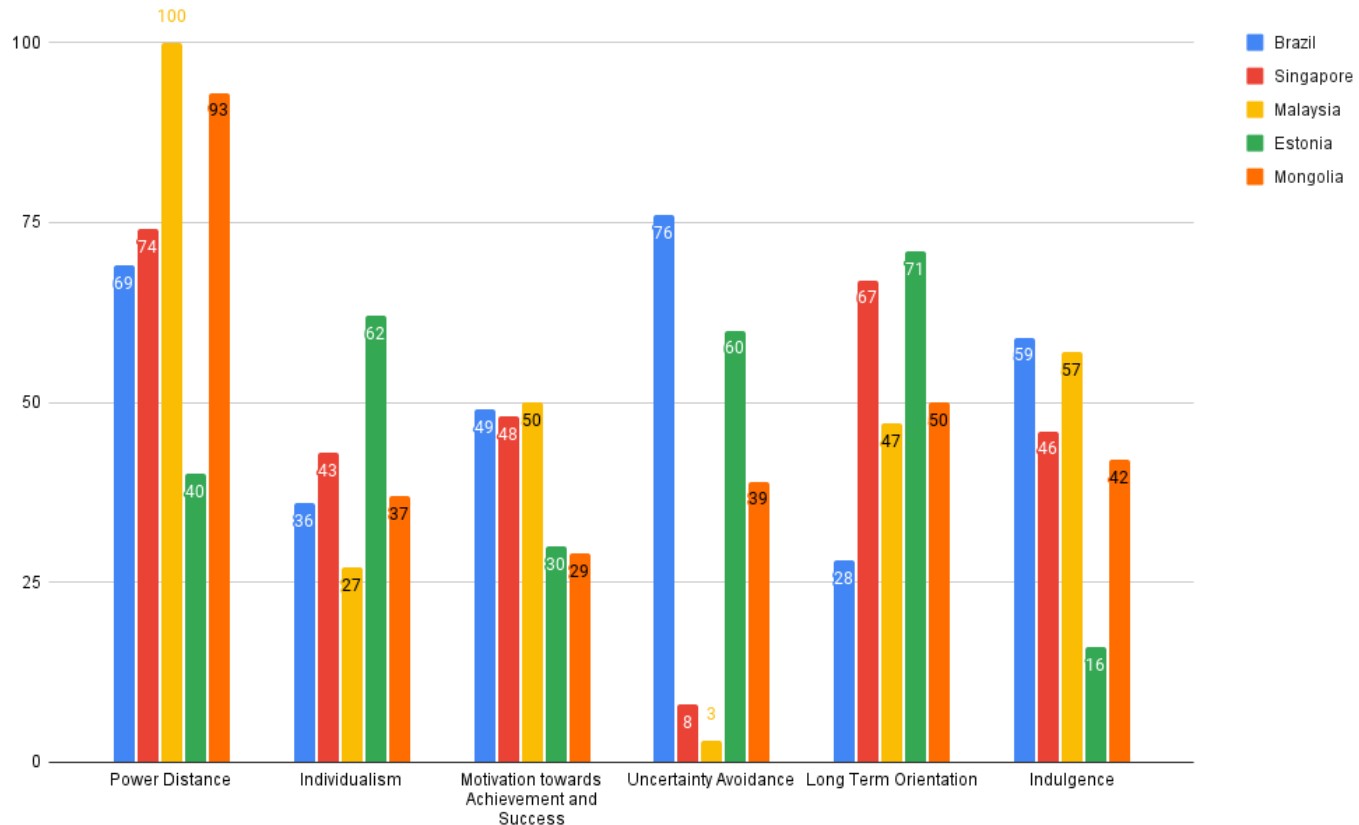

**Figure A1.** Full country comparison based on the cultural dimensions model.

## Appendix B. Measurement Model

A summary of the measurement model for the original model is presented in Table A1. The results of the assessment of DV for the refined model are presented in Table A2.

**Table A1.** Summary of measurement model evaluation results—original model.

| Item | Construct | Reliability (>0.5) | | | | | AVE (>0.5) | | | | | CR (>0.7) | | | | |
|---|---|---|---|---|---|---|---|---|---|---|---|---|---|---|---|---|
| | | BR | SIN | MAL | EE | MON | BR | SIN | MAL | EE | MON | BR | SIN | MAL | EE | MON |
| BEN1 | | 0.720 | 0.752 | 0.727 | 0.789 | 0.726 | | | | | | | | | | |
| BEN2 | BEN | 0.838 | 0.750 | 0.721 | 0.793 | 0.733 | 0.771 | 0.722 | 0.661 | 0.744 | 0.693 | 0.855 | 0.808 | 0.787 | 0.864 | 0.788 |
| BEN3 | | 0.754 | 0.663 | 0.534 | 0.649 | 0.620 | | | | | | | | | | |
| COM1 | | 0.496 ** | 0.725 | 0.627 | 0.688 | 0.567 | | | | | | | | | | |
| COM2 | COM | 0.729 | 0.780 | 0.704 | 0.742 | 0.741 | 0.623 | 0.656 | 0.678 | 0.612 | 0.682 | 0.736 | 0.783 | 0.773 | 0.710 | 0.810 |
| COM3 | | 0.644 | 0.463 ** | 0.704 | 0.406 ** | 0.737 | | | | | | | | | | |
| PR1 | | 0.692 | 0.656 | 0.923 | 0.688 | 0.630 | | | | | | | | | | |
| PR2 | PR | 0.605 | 0.641 | 0.313 ** | 0.836 | 0.000 ** | 0.707 | 0.693 | 0.530 | 0.779 | 0.401 ** | 0.835 | 0.936 | 0.227 ** | 0.870 | −0.138 ** |
| PR3 | | 0.824 | 0.781 | 0.355 ** | 0.813 | 0.573 | | | | | | | | | | |
| SA1 | SA | 0.840 | 0.793 | 0.828 | 0.851 | 0.619 | 0.852 | 0.811 | 0.846 | 0.827 | 0.735 | 0.831 | 0.773 | 0.828 | 0.802 | 0.756 |
| SA2 | | 0.865 | 0.830 | 0.865 | 0.803 | 0.852 | | | | | | | | | | |
| TR1 | | 0.658 | 0.623 | 0.700 | 0.735 | 0.841 | | | | | | | | | | |
| TR2 | TR | 0.797 | 0.786 | 0.654 | 0.796 | 0.805 | 0.756 | 0.738 | 0.700 | 0.775 | 0.855 | 0.852 | 0.834 | 0.816 | 0.857 | 0.919 |
| TR3 | | 0.813 | 0.805 | 0.747 | 0.795 | 0.918 | | | | | | | | | | |

** Values below the threshold.

**Table A2.** Fornell–Larcker criterion for refined model.

| | | BEN | COM | PR | SC | Trust |
|---|---|---|---|---|---|---|
| **BEN** | BR | **0.878** | | | | |
| | SIN | **0.850** | | | | |
| | MAL | **0.813** | | | | |
| | EE | **0.862** | | | | |
| | MON | **0.832** | | | | |
| **COM** | BR | 0.528 | **0.841** | | | |
| | SIN | 0.446 | **0.903** | | | |
| | MAL | 0.431 | **0.876** | | | |
| | EE | 0.425 | **0.891** | | | |
| | MON | 0.564 | **0.885** | | | |
| **PR** | BR | −0.408 | −0.216 | **0.894** | | |
| | SIN | −0.306 | −0.250 | **0.856** | | |
| | MAL | −0.237 | −0.044 | **0.807** | | |
| | EE | −0.534 | −0.263 | **0.889** | | |
| | MON | −0.105 | 0.130 | **0.869** | | |
| **SC** | BR | 0.606 | 0.330 | −0.352 | **0.923** | |
| | SIN | 0.639 | 0.453 | −0.403 | **0.901** | |
| | MAL | 0.588 | 0.405 | −0.330 | **0.920** | |
| | EE | 0.563 | 0.419 | −0.617 | **0.909** | |
| | MON | 0.495 | 0.482 | −0.051 | **0.857** | |
| **Trust** | BR | 0.708 | 0.481 | −0.522 | 0.622 | **0.869** |
| | SIN | 0.556 | 0.642 | −0.399 | 0.638 | **0.859** |
| | MAL | 0.646 | 0.406 | −0.255 | 0.615 | **0.837** |
| | EE | 0.608 | 0.607 | −0.576 | 0.730 | **0.880** |
| | MON | 0.722 | 0.555 | −0.219 | 0.730 | **0.925** |

The values in bold indicate the highest correlation of each item. In all cases, the highest correlation is with the same construct, demonstrating there are no issues with DV.

## Appendix C. Path Coefficients

Table A3 presents the bootstrapped *p*-values for the path coefficients.

**Table A3.** Summary of two-tailed *p*-values for path coefficients, obtained from bootstrapping in paired comparisons.

| | *p*-values | | | | | | | | | |
|---|---|---|---|---|---|---|---|---|---|---|
| | BRxEE | BRxMAL | BRxMON | BRxSIN | EExMAL | EExMON | EExSIN | MALxMON | MALxSIN | MONxSIN |
| BEN → Trust | 0.066 | 0.793 | 0.813 | 0.056 | **0.022** | 0.05 | 0.818 | 0.984 | **0.022** | **0.042** |
| COM → Trust | 0.067 | 0.726 | 0.891 | **0.011** | **0.047** | 0.097 | 0.355 | 0.854 | **0.008** | **0.021** |
| PR → Trust | 0.321 | **0.046** | 0.336 | 0.208 | 0.344 | 0.86 | 0.816 | 0.235 | 0.469 | 0.664 |
| SC →Trust | 0.167 | 0.649 | 0.076 | 0.658 | 0.478 | 0.625 | 0.403 | 0.278 | 0.962 | 0.223 |

Values in bold for significant differences (*p* < 0.05).

## Appendix D. HCTS Questionnaire

Table A4 presents the HTCS questionnaire revised based on this article's results. It comprises the constructs Competence (COM), Benevolence (BEN), Perceived Risk (PR) (reversed), and Structural Assurance (SA). The content within brackets should be adjusted according to the technology assessed.

**Table A4.** Revised HCTS.

| Construct | Item |
|-----------|------|
| COM1 | [Facial recognition systems] are competent and effective in [identifying dangerous individuals]. |
| COM2 | [Facial recognition systems] perform their role in [identifying potentially dangerous individuals] very well. |
| PR1 | There could be negative consequences when using [facial recognition systems]. |
| PR2 | It is risky to interact with [facial recognition systems]. |
| BEN1 | [Facial recognition systems] will act in my best interest. |
| BEN2 | [Facial recognition systems] will do their best to help me if I need help. |
| BEN3 | [Facial recognition systems] are interested in understanding my needs and preferences. |
| SA1 | I feel assured that legal and technological structures provided by the government protect me when using [facial recognition systems]. |
| SA2 | I can trust [facial recognition systems] because [Artificial Intelligence] systems are robust and safe. |

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
