# Peer review of "Assessing the Measurement Invariance of the Human–Computer Trust Scale"

_electronics, doi:10.3390/electronics14091806_

Round 1
Reviewer 1 Report
Comments and Suggestions for Authors
The paper's strengths lie in its clear motivation, comprehensive literature background, and detailed methodological execution. The theoretical grounding of the HCTS constructs is sound, and the explanation of the model’s adaptation to AI-based contexts is thoughtful and well-cited.
However, the study only achieved partial measurement invariance, meaning the HCTS cannot be reliably compared across most country pairs. While the authors acknowledge this and adopt an exploratory approach, the results must be interpreted with caution. More nuanced approaches to culture (e.g., Schwartz or GLOBE models) could offer deeper insight.
Sampling is another limitation. The use of convenience samples, along with imbalanced gender and age distributions, particularly in Singapore, likely affects the generalizability of the results. Though the authors address this in the limitations, it would be stronger if discussed earlier in the methods section.
Finally, the data presentation is sometimes overwhelming. Several MICOM tables are dense and could benefit from visual simplification or clearer synthesis in the discussion.
Reviewer 2 Report
Comments and Suggestions for Authors
This paper studies measurement invariance and its cross-cultural impact in the study of trust in human-computer interaction. Suggestions are as follows:
1. Please explain the theoretical connection between power distance, individualism, uncertainty avoidance and technology trust.
2. Provide more details of the questionnaire and video content, and the reasons for choosing facial recognition systems as a test scenario in law enforcement.
3. Summarize the significance of this study, such as how to use the results in a cross-cultural context, or provide specific suggestions for HCI practitioners in diverse groups.
Reviewer 3 Report
Comments and Suggestions for Authors
The article "Assessing the Measurement Invariance of the Human-Computer Trust Scale" investigates the validity and consistency of the Human-Computer Trust Scale (HCTS) across five countries: Brazil, Singapore, Malaysia, Estonia, and Mongolia. Employing the Measurement Invariance of Composite Models (MICOM) method, the authors evaluate the psychometric stability of the HCTS and discuss the implications of cultural differences in trust measurement.
One of the primary shortcomings lies in the theoretical grounding in the introduction and literature review sections. While the introduction correctly identifies the importance of measurement invariance and cultural influences on trust in technology, the authors inadequately articulate how their work uniquely contributes to the existing body of knowledge. The literature review extensively covers trust theories and prior validation studies; however, it predominantly offers descriptive summaries rather than critical comparisons or theoretical integrations. A more explicit discussion of how the current study significantly advances the theoretical understanding of trust across cultures would enhance the paper's clarity and purpose.
The methodology section, though detailed, also exhibits notable weaknesses. The authors' decision to utilize convenience sampling in each country is briefly mentioned but inadequately justified. Given the cultural nuances central to their argument, relying on convenience samples may significantly limit the representativeness and generalizability of their findings. Furthermore, the choice of using facial recognition systems as the technological focus of trust measurement is not sufficiently substantiated. The paper lacks a robust rationale for this specific context or a discussion regarding the potential biases or societal attitudes toward facial recognition technologies in each participating country. Addressing these concerns with more rigorous methodological justifications and contextual analysis would strengthen the study's validity and interpretative power.
Another critical issue arises from the authors' analytical treatment of the MICOM procedure. Although the MICOM method is robust, the paper inadequately addresses cases of partial invariance. The decision to proceed with analysis despite not achieving full measurement invariance is weakly justified, and the implications of this methodological compromise are not sufficiently explored. The authors repeatedly state that their analysis should be interpreted cautiously; however, they do not adequately discuss the potential consequences of partial invariance for interpreting cross-cultural comparisons. A deeper, more critical discussion on the analytical and interpretive challenges posed by these methodological limitations would enhance transparency and rigor.
The results and discussion sections also fall short of critically interpreting the outcomes. The authors identify certain constructs and items as problematic in specific samples, yet their interpretation lacks meaningful depth. For instance, despite removing problematic items from the scale, the authors do not critically reflect on whether these issues indicate fundamental conceptual problems with the HCTS or merely contextual discrepancies in interpretation across cultures. Furthermore, while interesting, the author's interpretation of the identified country groupings is overly speculative and insufficiently supported by empirical evidence or rigorous cultural analysis. A more thorough exploration of why certain constructs behave differently in specific national contexts, supported by qualitative or contextual insights, would significantly enrich the discussion.
Lastly, the paper underemphasizes ethical considerations regarding the cross-cultural measurement of trust and the use of facial recognition technologies in the context of law enforcement. Given that the technology under study (facial recognition) is ethically charged and controversial, especially concerning privacy, surveillance, and human rights, the lack of a robust ethical discussion is a significant oversight. Addressing these ethical dimensions explicitly would balance the methodological and analytical discussions and strengthen the paper's overall societal relevance.
Round 2
Reviewer 3 Report
Comments and Suggestions for Authors
The justifications added by the authors after the comments make the scope and contributions of their article clearer. Thus, it seems to be in an acceptable state for final publication.